# Mesoscale Models for Describing the Formation of Anisotropic Porosity and Strain-Stress Distributions during the Pressing Step in Electroceramics

**DOI:** 10.3390/ma15196839

**Published:** 2022-10-01

**Authors:** Radu Stefan Stirbu, Leontin Padurariu, Fereshteh Falah Chamasemani, Roland Brunner, Liliana Mitoseriu

**Affiliations:** 1Dielectrics, Ferroelectrics & Multiferroics Group, Faculty of Physics, Alexandru Ioan Cuza University of Iasi, 700506 Iasi, Romania; 2Materials Center, Leoben Forschung GmbH, 8700 Leoben, Austria

**Keywords:** porous ceramic, strain-stress, finite element calculations

## Abstract

Porous ceramics are often produced by using pyrolisable additives to generate porosity during the sintering step. The examination of the experimental microstructures of the resulted porous ceramics revealed certain levels of anisotropy, even if the original soft additives used as pore forming agents were spherical. The paper shows that anisotropic porosity may result in ceramics when using equiaxed soft polymeric additives for generating porosity, due to the deformation of soft inclusions during the pressing step. It has been found, by means of analytical and numerical calculations, that uniaxial pressing of a mixture of solid particles with contrasting mechanical properties (hard/soft) generates modifications in the shape of the soft phase. As a result, anisotropic shape distribution of the soft inclusions in the green ceramic body and elongated porosity in the final ceramic product are obtained. The elongated pores are statistically oriented with the major axes perpendicular to the pressing direction and will generate anisotropy-related functional properties. Analytical calculations indicate the deformation of a single soft inclusion inside a continuum solid. Further, by finite element simulations performed in 2D planes along the transversal and radial directions of the pressing axis, a bimodal angular distribution of the long axes of the soft inclusions has been found.

## 1. Introduction

Ceramic processing involves the cold compaction of powders to produce a green ceramic with a given shape and a certain degree of inter-grain cohesion (up to a relative density r.d. ~60%), followed by a subsequent thermal treatment at high temperatures (sintering step), allowing the densification of ceramics to high density values (>90% r.d.). The compaction process during uniaxial or isostatic pressing involves stress propagation both via rigid or flexible (die) walls and via the powder mass itself, as the particles contained in the powder distribute the stress by different kinematic processes involving sliding, rotation, particle deformation, rupture, etc. All of these processes are dependent on the starting powders (particle size, shape and distribution, their composition and their mechanical properties). Porous functional ceramics have been developed for a wide range of applications in gas sensing, thermal insulation, piezo- and pyroelectric harvesters, electrodes in solid oxide fuel cells, filtration elements, etc. [1,2]. They are composite materials containing an active matrix and isolated and/or interconnected air pores whose shape, size, distribution, anisotropy or multi-scale hierarchization play, together with the material properties of the dense matrix and the nature of the interfaces between them, an important role on the functional properties of the material as a whole. The porous ferro/piezo/pyroelectric ceramics can be prepared by various methods like incomplete sintering, by using pore forming agents (pyrolizable particles which are burned out by high temperature processing), or more sophisticated techniques as replica method, sacrificial phase technique, direct foaming method, paste extrusion, or the recent developed additive 3D printing [3,4]. The final size and pore shape, together with their orientation are determined by the type of sacrificial template materials (polymers, carbonaceous species, organic fillers, etc.) and by their deformability during the pressing step [5]. In a previous work [6], BaTiO_3_ ceramics with variable relative porosity levels (in the range of 8% to 26%) produced by using Poly(methyl methacrylate) (PMMA) spherical inclusions as sacrificial phase, have been analyzed. These ceramics were obtained by sintering of uniaxially pressed mixtures of oxide and polymeric powders. Even if it is clear that the final microstructures resulted after sintering are different from those of the green ceramics obtained after the pressing step (due to the processes of shrinkage, densification, grain growth, small porosity elimination, etc. taken place during the sintering), the elongated shape of the observed pores suggested that a plastic deformation of the soft equiaxed polymeric particles took place during the pressing.

The pressing step consists of an irreversible deformation transforming the material from a spare powder into a compact one (green ceramic). This process is characterized by a strong nonlinear relation between stress and strain which has been modelled in the literature by using three main approaches: (i) discrete element methods, which consider each powder particle as a rigid body without any plastic deformation [7,8], (ii) multi-particle finite element methods, taking into consideration the behavior and interaction of individual particles [9,10,11] or (iii) continuum finite element methods, where the powder is regarded as a mechanical continuum [12,13,14]. The density distribution, the shape of the green body and the crack formation during the pressing have been simulated by finite element methods [15,16] or by Drucker-Prager Cap constitutive models [17,18] whose reliability was validated by experimental procedures, as in ref. [19]. Such studies provided data concerning both the density and temperature distribution during the pressing step, and the final shape of ceramic body after sintering, crack formation, etc. [20,21]. 

There are no publications reporting pressing—induced shape modifications of the soft phase in a powder mixture formed by materials with contrasting elastic properties (e.g., composed by a hard and a soft component). Therefore, a simplified analytical and a numerical approach have been developed in the present paper for describing the strain-stress fields and to observe the deformation of the soft inclusions (e.g., PMMA spheres) into the hard matrix (oxide powder) during the pressing step of a composite powder. Intuitively, a longer bond of a body is easily deformable than a shorter one. This might be the cause of a tendency of the pores to elongate along the axis of the cylindrical sample, as observed in ref. [6]. This represented the starting point of the idea of studying the deformation of the soft inclusions in a bulk matrix, while applying pressure and observing how those deformations might induce anisotropy, if any. Thus, from a fundamental perspective, this study might be interesting for understanding the relationship between local properties (soft-hard interfaces), final microstructures and the functional properties, further finding local strain-stress distributions and for the design of specific microstructures in order to produce enhanced functional properties in any type of ceramic, irrespective of composition. Therefore, the present study proposes to describe, by a theoretical and a modelling approach, the role of the uniaxial and isostatic pressing on the deformation of mechanically soft phase introduced as a sacrificial template (e.g., PMMA polymeric spheres) in the ceramic powders in order to provide porosity.

## 2. Materials and Methods

Barium titanate ceramics with variable porosity degree were produced by adding to the starting BaTiO_3_ oxide powders monodisperse Poly(methyl methacrylate) (PMMA) microspheres with diameters of ~10 μm (Figure 1a). The mixed powders have been coldly pressed at 150 MPa (uniaxially or isostatic) and then, the green ceramics have been densified by applying a pressureless multi-step thermal treatment to ensure the combustion of polymeric particles, the complete gas elimination before closing the ceramic pores and complete sintering, as described in detail in ref. [6]. The surface ceramic morphology has been examined in fresh ceramic fractures by high resolution scanning electronic microscopy (SEM) with a Carl Zeiss System NEON40EsB. In order to better observe the 3D pore morphology in the sample, 3D micro X-ray computed tomography (μ-XCT) [22,23,24] with a nanotom m from General Electrics (GE) was pursued. The voxel size was set to 1.33 µm and the selected volume of interest (VOI) for the analysis was 186 × 186 × 186 μm^3^ big. Python libraries (including Numpy 1.18.1, Scipy 1.4.1, Scikit-image 0.16.2, Matplotlib 3.1.3, and Scikit-fmm 2019.1.30) and Avizo^®^ ThermoFischer Scientific, version 2019.1 and 2021.2 have been used to perform the image processing and visualization of the µ-XCT data. In order to compute the strain-stress fields and deformation of such structures, containing a maximum amount of 950 equiaxed soft inclusions in a hard matrix, some numerical methods based on the commercial finite element codes (COMSOL Multiphysics v. 6.0, COMSOL Inc., Burlington, VT, USA) have been used. The statistical analysis to determine the angular distribution and deformability level, as result of the isostatic and uniaxial pressing was performed by using C/C++ programs.

## 3. Results and Discussion

### 3.1. Microstructural Characterization

In the case of uniaxial pressing, irrespective of the concentrations of spherical polymeric addition shown in Figure 1a, one can observe that in the ceramic microstructures the resulting pores after burning out the polymeric additive have the tendency to be rather elongated (Figure 1b) than spherical [6]. A slightly elongated shape of the pores is also generated when using isostatic pressing. The shape of such pores is almost equiaxed with respect to the case of uniaxial pressing (Figure 1c). For comparison, the intrinsic ceramic porosity resulted in ceramics by incomplete sintering (without any polymeric addition) shows quasi-spherical isotropic porosity (Figure 1d). 

A complementary microstructural characterization of the sintered ceramics realized by using the micro X-ray computed tomography (μ-XCT) provided 3D images which can give a more detailed information concerning the volume distribution of porosity and of its possible anisotropy, at the available resolution limit. The obtained μ-XCT data was pre-processed in the local coordinate system, according to the following procedure: (i) importing of the raw data from the μ-XCT analyses in a (140 × 140 × 140) array; (ii) assign to a special position the values +1, if it is part of a pore and −1, in case it belongs to a bulk region; (iii) upscaling to a system of (280 × 280 × 280) size (each initial voxel was divided into other 8) and using a smoothing algorithm in order to correct some voxels at interfaces; (iv) smoothing the interfaces of the represented images, performed by slight modifications of the coordinates of the nodes belonging to the interfaces. The resulting 3D images of the processed structures derived from the BaTiO_3_ ceramics with a relative porosity of about 19%, uniaxially pressed (Figure 2a) and with a relative porosity of about 13%, isostatically pressed (Figure 2b) indicate the presence of certain anisotropy resulted by both types of pressing, when using spherical sacrificial polymeric additions.

In order to detect the presence of anisotropy along a given axis, correlation functions were used, as defined by the relations:(1)$$\begin{array}{c} C_{x}\left(\Delta x\right)=\langle V\left(x,y,z\right)\cdot V(x+\Delta x,y,z)\rangle -\langle V(x,y,z)\rangle \cdot \langle V\left(x+\Delta x,y,z\right)\rangle \\ \;C_{y}\left(\Delta y\right)=\langle V\left(x,y,z\right)\cdot V(x,y+\Delta y,z)\rangle -\langle V\left(x,y,z\right)\rangle \cdot \;\langle V(x,y+\Delta y,z)\rangle \\ C_{z}\left(\Delta z\right)=\langle V\left(x,y,z\right)\cdot V(x,y,z+\Delta z)\rangle -\langle V(x,y,z)\rangle \cdot \;\langle V(x,y,z+\Delta z)\rangle \end{array}$$
where V(x,y,z) are the values assigned to the voxels: $$V\left(x,y,z\right)=\{\begin{array}{c} 1,\;for\;pores\\ -1,\;for\;bulk \end{array}$$

The dependences of such spatial correlation functions on the distance between randomly chosen voxels, determined for the two types of 3D microstructures along the three local axes are presented in Figure 2c,d. In both cases the anisotropy is detected since the three plots along the main axes do not overlap. For the uniaxial pressing, if the local coordinate systems axes match the macroscopic coordinates (i.e., the pressing direction and two others belonging to the normal plane to the pressing direction), then one can expect an overlapping of the correlation plots along two directions with the third one along the pressing direction having significantly smaller values. Figure 2c shows that the Oy direction corresponds to minimum correlation values, but the other two plots along Ox and Oz directions are not fully overlapping, even if they are quite close to each other. This means that the local Oy direction is close, but not coincident to the macroscopic pressing direction of this sample. Nevertheless, the fact that this direction is characterized by the smallest correlation length of 3.5 μm suggests that a flattening of the soft polymeric inclusions along the pressing direction takes place. The 3D microstructure resulted for the isostatically pressed ceramic presented in Figure 2b is similarly analyzed by means of correlation functions shown in Figure 2d. In this case, one can distinguish that the local coordinate system perfectly matches the main anisotropy axes, since the correlation plots along Oy and Oz overlap, while along the Ox direction takes smaller values. In the same time, the longer correlation length identical on the Oy and Oz axes suggests a preferential pore elongation along these two perpendicular directions. This analysis indicates that both types of pressing provide a degree of pore anisotropy within the ceramic volume, but the shape of pores in the uniaxially pressing changes more with respect to the original spherical one, as indicated by the existence of a minimum correlation length of 3.5 μm.

In conclusion, the microstructural analyses indicated a certain degree of pore anisotropy and an elongated pore shape which seems to be more pronounced for the ceramic consolidated by uniaxial pressing. It is known that the final microstructure in a presureless sintered ceramic is different than one of the started green body due to the shrinkage accompanying the densification and grain growth. However, the initial shape and distribution of the soft polymeric phase within the oxide powder matrix from the green ceramic is expected to play a predominant role on the final pores size, shape and distribution in the sintered body. Therefore, a theoretical and a modelling approach are proposed in order to describe the role of the uniaxial and isostatic pressing on the deformation of mechanically soft phase used as sacrificial template to produce porosity. First of all, the deformation during the pressing of a single equiaxed soft inclusion placed in various positions inside a continuum rigid matrix is described by an analytical approach, based on the elasticity theory, thus allowing to determine both the stress-strain fields in various positions around the inclusion and the deformation of its boundaries. Further, finite element numerical methods are proposed to determine the strain-stress fields in the composite powders containing equiaxed soft inclusions into a rigid matrix. Their distribution over the angular orientation and degree of deformability have been determined and discussed as result of the isostatic and uniaxial pressing.

### 3.2. Modelling

#### 3.2.1. Analytical Approach

In order to build a simple mathematical model, capable of delivering some consistent information about the investigated pressing process of a mixture of soft and hard composite powders, the following simplifications were proposed: (1) the powder mixture is introduced in an infinitely long cylinder, in order to avoid the boundary effects produced at the top and bottom plane surfaces of the cylinder, together with the circular edges, on their adjacent volumes; (2) one single spherical soft inclusion is taken into consideration (this approximation can describe the situation of small PMMA concentrations, so that the boundaries separating the hard/soft materials will not interact to each other); (3) the stiffness of the polymeric inclusion is negligible comparing to the one of hard ceramic powder; (4) the overall hard phase outside the soft sphere inclusion is a continuous homogeneous and isotropic material.

The effective mechanical characteristics of the hard matrix were firstly estimated in order to determine the role of the pressing step of the powder mixture formed by soft and hard particles on the resulted anisotropy of a unique soft inclusion. One should consider that the matrix outside the soft inclusion is not a full densified material, but it should be described by effective mechanical properties characterizing a porous structure. For this aim, mixing approaches based on Voronoi structures were employed and analytic formulae for the effective Young’s modulus and Poisson’s ratios were derived [25,26] as described in Appendix A.

While the inner radius is a constant, the external radius $\vec{R_{e}}$ is vectorially determined as shown in Figure 3c. The stress Equations were further derived by separating an elementary volume in which normal (longitudinal) $\sigma_{l}$, radial $\sigma_{r}\;$ and tangential $\sigma_{t}\;$ stresses are indicated (Figure 3a). The normal stress is calculated by applying Hooke’s law in each slice of the cylinder containing a soft inclusion, when pressure is applied to its basis (Figure 3b):(2)$$\sigma_{l}=\frac{p\pi R_{e}{}^{2}}{\pi \left(R_{e}{}^{2}-R_{i}{}^{2}\right)}$$
where p is the pressure applied on the plane surfaces, $R_{i}$ is the radius of the soft inclusion and $R_{e}$ is the external radius of cylinder, while the radial and tangential stresses (Figure 3d) satisfy the equilibrium condition on vertical direction: (3)$$r\sigma_{r}d\theta +2dr\sigma_{t}\sin\left(\frac{d\theta }{2}\right)-\left(r+dr\right)\left(\sigma_{r}+d\sigma_{r}\right)d\theta =0$$

After some mathematical manipulations, the final form of the stress Equation becomes:(4)$$\{\begin{array}{c} \sigma_{r}=\frac{R_{i}^{2}p_{i}-R_{e}^{2}p_{e}}{R_{e}^{2}-R_{i}^{2}}-\frac{R_{i}^{2}R_{e}^{2}\left(p_{i}-p_{e}\right)}{R_{e}^{2}-R_{i}^{2}}\cdot \frac{1}{r^{2}}\\ \sigma_{t}=\frac{R_{i}^{2}p_{i}-R_{e}^{2}p_{e}}{R_{e}^{2}-R_{i}^{2}}+\frac{R_{i}^{2}R_{e}^{2}\left(p_{i}-p_{e}\right)}{R_{e}^{2}-R_{i}^{2}}\cdot \frac{1}{r^{2}} \end{array}$$

In order to compute the radial $u_{r}$ displacement on the pore’s boundary, when the system is subjected to isostatic pressure, the following Equation is used: (5)$$u_{r}=r\cdot \frac{1-\nu }{E}\cdot \frac{R_{i}^{2}p_{i}-R_{e}^{2}p_{e}}{R_{e}^{2}-R_{i}^{2}}+\frac{1}{r}\cdot \frac{1+\nu }{E}\cdot \frac{R_{i}^{2}R_{e}^{2}\left(p_{i}-p_{e}\right)}{R_{e}^{2}-R_{i}^{2}}$$

A suite of C/C++ programs has been written to compute the stress-strain fields in various positions around the soft inclusion, whose axis will be considered as a reference for the Equations, and to determine the way that the boundary of the inclusion deforms. As a first step, the program computes the effective values of the external cylinder radius and, then, it applies the Equations (2) and (4)–(6), in order to determine the equivalent stress and the deformed shape of the elemental boundary. The von Mises stress derived from the modern theory of the maximum elastic potential energy [27] represents a mathematical tool that replaces all the main and secondary stresses accumulated in the material, by a singular quantity that would introduce the same energy into the body, thus, creating similar breaking effects. The formula used for the estimation of the equivalent von Mises stress is given by the relationship:(6)$$\sigma_{vonMises}=\sqrt{\sigma_{l}{}^{2}+\sigma_{t}{}^{2}+\sigma_{r}{}^{2}-\sigma_{l}\sigma_{t}-\sigma_{t}\sigma_{r}-\sigma_{l}\sigma_{r}}$$

A representative image (obtained by exporting the graphic files to MeshLab) of the deformed soft inclusions placed in various positions with respect to the cylinder axis is presented in Figure 4, where the parameters: $R_{e}\;$ = 5 mm, $R_{i}\;$ = 0.5 mm, $\;p_{e}\;$ = 150 MPa, $p_{i}\;$ = 0 MPa, $E_{m}=$ 2000 MPa, ν  = 0.25 were used. A detailed representation is shown in Appendix A. The distribution of the von Mises stress accumulated in the material just around the surface of the soft inclusion is indicated in a colour scale (the green sides represent the lowly-stressed areas, while the red ones represent highly-stressed regions). The general tendency of the deformed inclusions (initially spherical) is to take an ellipsoid-like shape, no matter their radial position, which seems to be realistic, as it was observed in the ceramic microstructures (Figure 1b). The von Mises stress behaviour however, tends to vary considerably from a highly symmetric distribution when the inclusion is placed on the cylinder axis, up to a state where all the significant stresses tend to accumulate on a narrower strip pointing out to the exterior boundary of the system, up to such a concentration that it squeezes the material inside the hollow sphere (Figure 4). This approach allowed to make an interesting observation: the inclusion located just in the vicinity of the cylinder boundary (Figure 4d) displays, besides the elongation, a tendency of longitudinal squeezing. This suggests that the interactions between geometrical boundaries play an important role on the local strain-stress fields.

The present simplified approach provides qualitative results if the ratio $R_{e}/R_{i}\;$ > 500, when the stress calculation becomes inaccurate, due to a program limitation related to the machine power of approximation. This limits the possibility of mimicking a real setup because, for a 10 mm holder, the smallest usable inclusion radius is of around 0.03 mm, while a real inclusion has the radius in the order of 10^−8^ mm. Other important limitations are determined from the mathematical apparatus itself considering for calculations a single soft inclusion and from the effects of the top and bottom flat surfaces on the nearby material, which cannot be determined by using only the Hooke’s law. Another drawback is determined by the impossibility of studying the interaction between more soft inclusions and their geometrical boundaries. Overall, those results are promising because the deformed shape of the outermost inclusion matches the observed shape of a squeezed hollow sphere near the boundary of a solid cylinder, but these overall observations suggest that a numerical model would be more appropriate.

#### 3.2.2. Numerical Approach by Finite Element Modelling

In order to simplify the study, instead of a 3D approach, 2D analyses were further performed in both longitudinal and transversal section of the cylinder, thus, from this point on, the inclusion shall be referred as circular. Such a numerical model would describe correctly the behaviour in the transversal section, which represents a reduction of phenomena taking place in parallel plans, which do not mechanically interfere much since the differences in the behaviour of two adjacent parallel plans may cause only the apparition of secondary stresses. Some limitations of the proposed approach are expected in the longitudinal section, in which radially disposed planes intersect much often, by means of both main and secondary stresses. Nevertheless, such analysis was chosen here for its simplicity.

As a first test, the results of the numerical approach were qualitatively compared for the case of a single soft inclusion with similar size to the ones resulted from the analytical calculations. Appendix A comparatively present the deformation and von Mises stresses in longitudinal and transversal sectioning plans determined by the analytical and numerical method. Even if the von Mises stress distributions do not fully match due to the effects of the secondary stresses that were not considered in the analytical approach, the deformed shape of the inclusion resulted from the two approaches matches fairly well and the numerical approach will be further used for a larger number of soft inclusions.

***A.*** 
**
*Isostatic pressing*
**


The deformation of the soft circular inclusions inside a more rigid matrix during the isostatic pressing step was simulated by using COMSOL Multiphysics platform. In a first place, a rectangle with the aspect ratio of 3:1 is filled with randomly placed hollow circles that mimic the soft inclusions, so that the hollow area of the whole system remains around ~30%. To increase the accuracy and insure the best statistical relevance, larger systems, with randomly-generated initial structures, are gradually built (Figure 5a), with increased number of such inclusions, up to 950 (COMSOL Multiphysics does not support more than 1000 geometrical entities in a single model), while keeping as constant the aspect ratio of 3:1, the radii of the inclusions of 0.5 mm and the lack of percolation. After the system is created, another very small inclusion is placed in the center of the rectangle, acting as a fixed anchor point. In the following calculations, several randomly generated (using the same governing parameters) structures have been studied. Since the structures are very similar, the number of elements sits around 15,000 and the number of nodes around 10,000. A meshing example, together with the boundary conditions are presented in Appendix A respectively.

As an input, the material constants: Young’s modulus of 100 GPa and Poisson’s ratio of 0.25 were used, together with boundary conditions: (i) fixed anchor at the centre of the rectangle and (ii) external pressure $p_{e}$ = 500 MPa on its edges. The mesh parameters were left to the decision of the software and several analyses have been performed. Figure 5b displays the deformed shapes of the structures presented in Figure 5a together with the von Mises equivalent stress by means of a colour scale (blue represents the minimum value, while red corresponds to the maximum stress). By these calculations, it clearly results that the edges of the inclusions influence very much the local displacements, which confirms one of the hypotheses of the study. Moreover, one can observe that clustering chains tend to form inside the material, which might dictate the behaviour to a greater extent than the exterior limits themselves.

In order to determine if some degree of anisotropy is generated after applying an isostatic pressure to the system, the angles between the longest diameter of each deformed inclusion and the horizontal axis were determined, as described in Appendix A and their distribution was plotted.

Figure 6 displays a broad bimodal angular distribution of the soft inclusions’ long axes for a number of 950 elements which suggests the presence of two possible main anisotropy directions approximatively placed at ±50°, not far away from being perpendicular. A higher degree of symmetry should be obtained when using a larger number of soft elements in simulations, for a better statistic. This result confirms the observation from Figure 2b,d in which one could identify the presence of two main perpendicular anisotropy axes. Further, the model building algorithm was modified to describe rounded structures in order to simulate the transversal section. Measuring the angle between the main diameter of an inclusion and the horizontal axis was not convenient in this case, because such a statistic would be hard to visualize and to get useful information out of it. Therefore, the analysis program has been adapted to calculate the angle between the main diameter of each inclusion and the line connecting its centre of mass with the section’s centre. The obtained angular distribution of the inclusions against a radial direction is exposed in Figure 7, which also displays the initial and deformed structure containing 850 soft inclusions in a cylinder with 24 mm diameter. It is observed that this distribution (Figure 7d) is close to a normal one and is centred around zero degree, thus indicating a clear preference for a purely radial orientation of the inclusions, in a transversal plane, which is a sign of a very good isotropy in the plane perpendicular to the cylinder axes. This demonstrates that even the starting soft fillers were equiaxed (as shown in Figure 1a), they deform as result of the uniaxial pressing and generate, after sintering, anisotropic porosity with elongated pores having the short axes along the pressing direction and their long axes in a plan transversal to it. As it was expectable, there is no identifiable anisotropy is the transversal section. In the longitudinal section however, slight anisotropic tendencies are observed. As already discussed, this general anisotropy is not an evident one and can be traced back to several more or less disputable sources. Even if, at this very moment, one cannot give a clear picture about the general anisotropy in the 3D structure, around particular regions, local anisotropy is clearly present and is strongly dependent on the clustering degree of the soft equiaxed inclusions used as sacrificial templates (PMMA circles) and this anisotropy would affect the functional properties (dielectric, ferro/piezo/pyroelectric, storage properties, etc.) of such a porous ceramic structure.

***B.*** 
**
*Uniaxial pressing*
**


For the case of uniaxial pressing, zero pressure was considered on the sides of the rectangle (Figure 8a). Intuitively, it feels natural that the rectangle had the tendency to shorten and the inclusions had the tendency to flatten with their main axis standing at an angle close to 90° degrees about the main axis of the cylinder (Figure 8b). Indeed, the computed structure presents anisotropy and deformed inclusions with the long axes perpendicular to the pressing (vertical) direction, as shown in Figure 8c and the corresponding angular distribution is bi-modal, with sharp maxima around the angle of ±90°, almost symmetrical, similar as observed in the SEM micrographs in Figure 1b.

In order to further evaluate the specific features related to the local deformations of the soft inclusions when using the two pressing methods, the sizes of the long (*R*) and respectively, of the short (*r*) axes have been computed for all the soft inclusions and their weights have been plotted as a function of their specific aspect ratio *R/r*. The obtained distributions presented in Figure 9 indicates a clear difference between the boundaries’ behavior of the soft inclusions after being subjected to the two types of pressing procedures. It is observed in the case of isostatic pressing the narrow distribution with a rather Gaussian aspect, with a sharp maximum around R/r ~ 1.3, thus indicating a high degree of shape homogeneity, even if the angular distribution of the main axes is broader and bimodal. Meanwhile the broader, log-normal type distribution corresponding to the uniaxial case, with a shallow maximum around R/r~2, showes a lesser gometrical homogeneity of the system, but a higher degree of anisotropy, due to the narrower, angular distribution, as shown in Figure 8d. These results qualitatively agree with the observations from Figure 2a,c in which more flattened pores were detected in the case of ceramics processed from uniaxially pressed powders, as resulted from a higher degree of deformation of the soft polymeric circles during the uniaxially pressing step.

The limitations of the present numerical approach are related to the simplifications and assumptions employed, like: the lack of percolation, the perfectly homogeneous and isotropic character of the material matrix, the perfectly shaped circular soft inclusions and possible mesh-related errors. One might argue that the linear-elastic material model is unsuitable to describe the behaviour of the powder matrix during the compaction process. In order to address this issue, a sequence of simulations with variable matrix material properties has been performed (Appendix A). Inspired by the Drucker-Prager model, the elastic modulus was modified at each step, according to the resulting von Mises stresses computed at the previous step. After a few computational steps, one can distinguish that the shape of the deformed inclusion is very similar to the one resulting from the constant material properties case. So, it can be concluded that the present approach is suitable for describing the behaviour of the body during the pressing process. It is worth to mention that the simplifications related to the use of a 2D approach provide valuable results describing systems with elongated soft inclusions (e.g., cylinders or very long ellipsoids) in a transversal section, rather than spherical soft inclusions. Other possible errors might be determined by the limited precision in the calculation of inclusions’ individual mass centres and in the base change of the whole structure.

## 4. Conclusions

Finding the relationship between the functional properties of electro-ceramics at different scales is highly challenging and multiscale analysis is necessary to complete the composition-nano/microstructure-properties picture. Among the microstructural factors, the ceramic density is important because first of all porosity is unavoidable during the processing and secondly, because specific porosity distribution determining specific strain-stress distributions may provide superior piezo/ferro/pyroelectric properties. In this paper, the role of the isostatic and uniaxial pressing step on the formation of anisotropic porosity in BaTiO_3_ ceramics when using as pore forming agent polymeric spherical particles is investigated. The presented model works equally well for any matrix material, as long as it can be considered elastic, homogeneous and isotropic. Using analytical and numerical calculations it is shown that during the pressing step of a mixture of solid particles characterised by contrasting mechanical properties (hard/soft), the shape of the soft deformable phase, initially equiaxed changes both when using uniaxial or isostatic pressing. In both cases, anisotropic distribution of such inclusions in the green body resulted. A higher shape homogeneity and a broad bi-modal angular distribution of the elongated inclusions along two perpendicular main axes resulted by simulations for isostatic pressing. In contrast, a broad log-normal distribution of the inclusion’s aspect ratio showing a stronger shape inhomogeneity and a high degree of anisotropy resulted for the uniaxial pressing. Starting with such green ceramic microstructures, the deformed inclusions would generate after sintering elongated and anisotropic porosity microstructures and anisotropic related functional properties. This type of porosity originated from the soft polymeric pyrolisable additives is different than the naturally occurred one in a sintered ceramic, as resulted by incomplete sintering (which is isotropic and pores are equiaxed). The first part of this study using an analytical approach for a single equiaxed inclusion into a continuum matrix was able to give information concerning the tendency of the soft phase to elongate along the symmetry axis, thus, resulting in a strong anisotropy in one direction. The finite element numerical approach performed in 2D planes along transversal and radial directions indicated the anisotropy of the soft phase along two main directions, symmetrical with respect to the major cylinder axis (i.e., bimodal distributions). For the two types of pressing, elongated soft inclusions are found, with different shape factor distributions and anisotropy. The numerical approach can be further improved by considering non-linear mechanical characteristics for the ceramic powders, coupling the elastic properties with the Drucker-Prager theory in order to better understand the consolidation step and by building a 3D model able to consider more elastic effects taking place in the bulk matrix.

## Figures and Tables

**Figure 1 materials-15-06839-f001:**
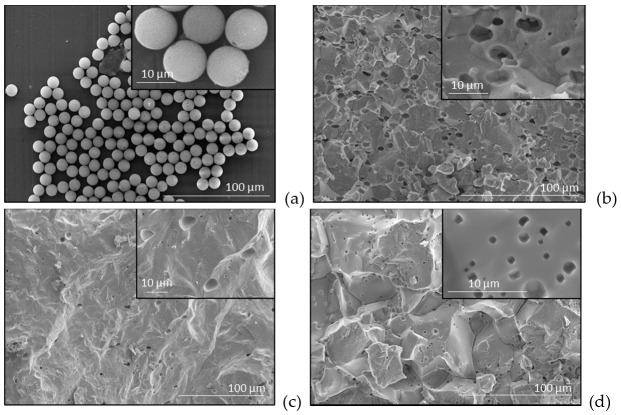
SEM micrographs in fractured fresh surfaces of the PMMA spheres used as pore forming agent (**a**) and for the sintered BaTiO_3_ ceramics with elongated porosity (~10 vol.%) obtained by the addition of PMMA spheres uniaxially pressed (**b**), or isostatically pressed (**c**) and for a BaTiO_3_ ceramic realized without any polymeric addition, showing natural occurred spherical porosity (4 vol.%), as result of imperfect sintering (**d**).

**Figure 2 materials-15-06839-f002:**
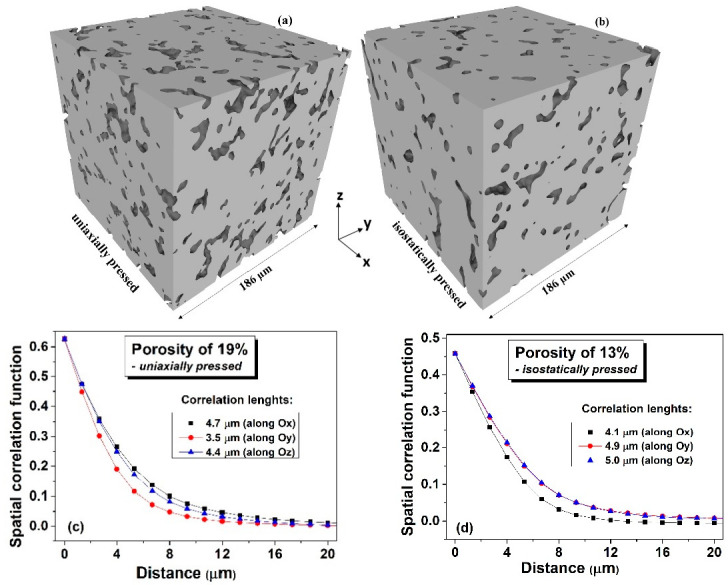
3D images of the processed BaTiO_3_ ceramics structures derived by micro X-ray computed tomography of with a relative porosity of ~19%, uniaxially pressed (**a**) and with a relative porosity of ~13%, isostatically pressed (**b**); (**c**,**d**) Correlation functions along the main local axes for the structures shown in the figures (**a**,**b**).

**Figure 3 materials-15-06839-f003:**
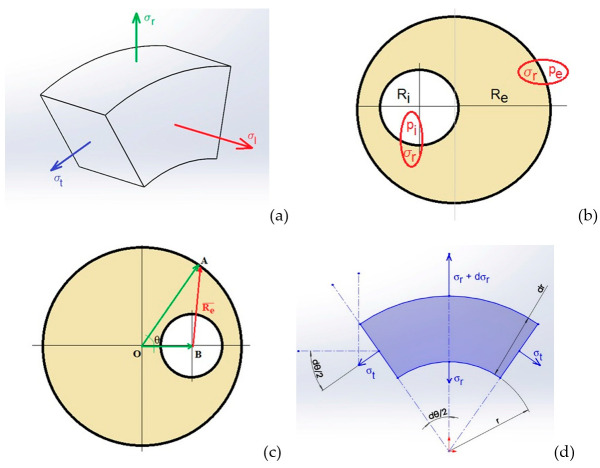
Illustration of the main stresses in a cylindrical body (**a**); radial and tangential stresses on an insulated volume element (**b**); calculation of the external radius: $\vec{R_{e}}=\vec{OA}-\vec{OB}$ (**c**); illustration of the mechanical equilibrium conditions (**d**).

**Figure 4 materials-15-06839-f004:**
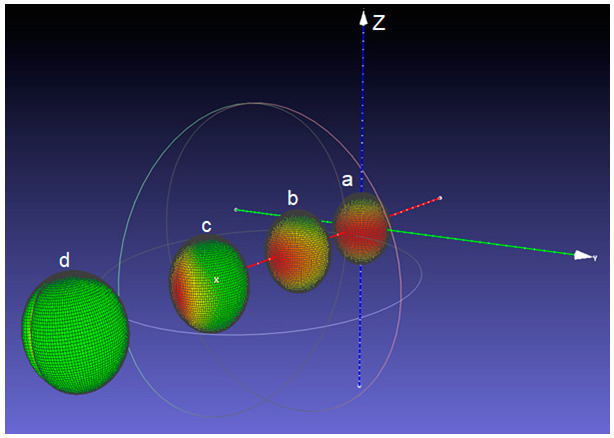
Position of the presented inclusions relative to the blue cylinder axis Oz: r = 0 (**a**), 1.47 mm (**b**), 2.93 mm (**c**) and 4.4 mm (**d**).

**Figure 5 materials-15-06839-f005:**
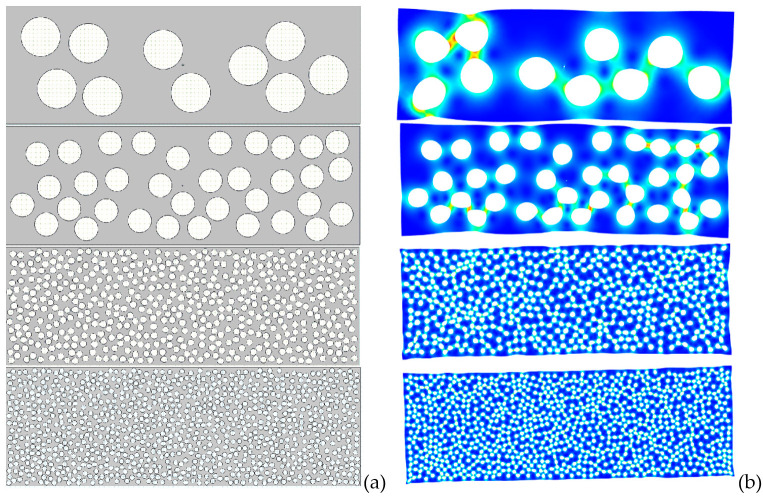
Systems with 10, 50, 500 and 950 circular inclusions (**a**) and the corresponding simulated microstructures after the deformation by isostatic pressing (**b**).

**Figure 6 materials-15-06839-f006:**
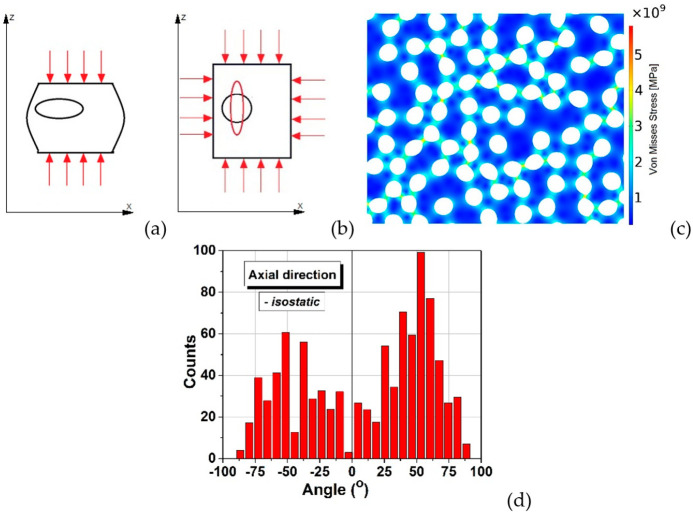
Schematic of isostatic pressing: (**a**) pressure applied only along the cylinder axis, (**b**) after isostatic pressing and deformation; (**c**) isostatically deformed structure; (**d**) angular distribution of inclusions derived for the case of isostatic pressing.

**Figure 7 materials-15-06839-f007:**
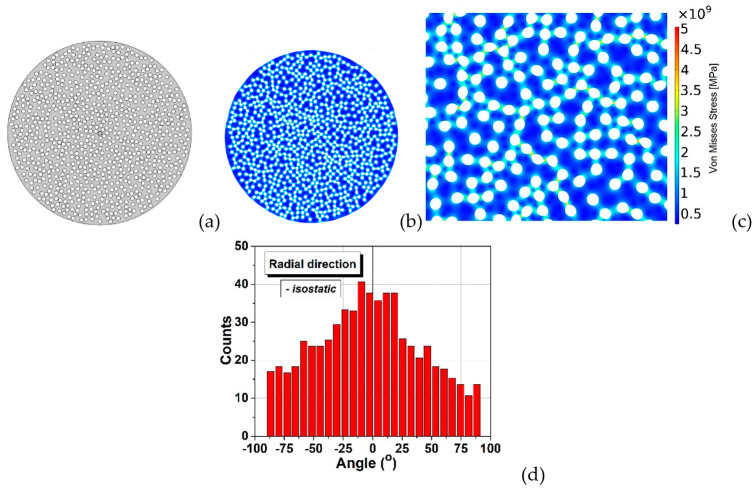
Initial (**a**) and deformed (**b**) shape of the soft inclusions in the transversal section for the case of isostatic pressing; (**c**) detail of the deformed structure and von Mises stresses; (**d**) statistical angular distribution in the transversal section.

**Figure 8 materials-15-06839-f008:**
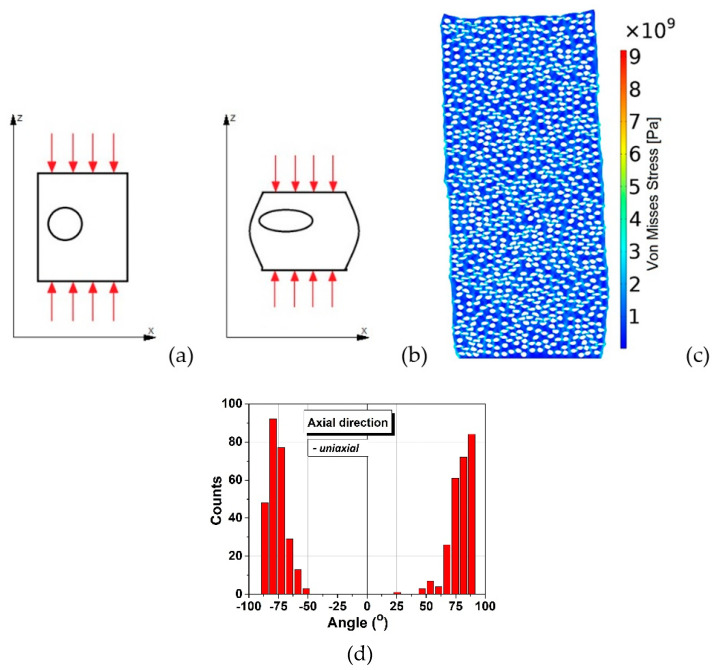
Schematic of uniaxial pressing: (**a**) before pressing, (**b**) after pressing and deformation; (**c**) uniaxially entirely deformed structure; (**d**) angular distribution of inclusions derived for uniaxial pressing.

**Figure 9 materials-15-06839-f009:**
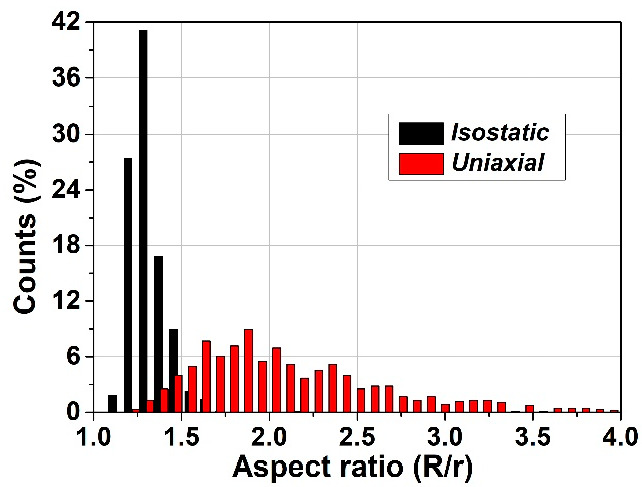
Comparative plot of the aspect ratio distributions of the soft phase for the isostatic and uniaxial pressing.

## Data Availability

Data are contained within the article. Any other supplementary information is available on request from the corresponding author.

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
