# Peer review of "Mesoscale Models for Describing the Formation of Anisotropic Porosity and Strain-Stress Distributions during the Pressing Step in Electroceramics"

_materials, 2022, doi:10.3390/ma15196839_

Round 1
Reviewer 1 Report
The written article needs a minor revision and the following questions are raised about this article:
The numerical simulation part should specify the number of element and nodes used in modeling, the application of boundary conditions and display the grid in in the form of image .
Reviewer 2 Report
The article presents an interesting study concerning the change in the shape of soft initially spherical inclusions in powdered ceramics under two types of pressing. This study shows that the shapes of the pores eventually become non-spherical. The work uses several assumptions that allow predicting the final porosity structure in piezoceramics only qualitatively. Thus, the problems are solved in a linear setting; finite element modeling is carried out for a two-dimensional problem, and not for a three-dimensional one; distortion of pore shapes during sintering is not taken into account, etc. Nevertheless, these obtained approximate results are new and represent a certain scientific value. Therefore, the work can be recommended for publication after taking into account the following remarks.
1) In the introduction, it is recommended to provide more references to review papers devoted to porous piezomaterials, such as:
Levassort F, Holc J, Ringgaard E, Bove T, Kosec M, Lethiecq M. Fabrication, modelling and use of porous ceramics for ultrasonic transducer applications. J. Electroceram. 2007, 19: 127–139. doi: 10.1007/s10832-007-9117-3
2) Also noteworthy are works showing the influence of pore shapes on the effective moduli of porous piezoelectrics, such as:
Banno H. Effect of shape and volume fraction of closed pores on dielectric, elastic and electromechanical properties of dielectric and piezoelectric ceramics - a theoretical approach. Amer. Ceram. Soc. Bull. 1987, 66(9): 1332-1337.
3) Designations in fig. 3(c) are poorly visible. Maybe the designations “t” and “r” for stresses should be subscripts?
4) It is not clear what $\sigma$ is without subscripts in formula (3). Probably there should also be subscripts.
5) The designations in Figure 3(b) must also have subscripts. In addition, it is not clear if the inclusion is shifted along the horizontal axis from the center, then why this shift does not affect the stresses in the analytical formulas.
6) Since two-dimensional problems were solved in the finite element analysis (probably, about plane strain, which, however, is not indicated explicitly), in this section we should not talk about spherical pores, but about cylindrical pores or circular pores.
Reviewer 3 Report
Manuscript Mesoscale models for describing the formation of anisotropic porosity and strain-stress distributions during the pressing step in electroceramics, Authors:
Radu Stefan Stirbu, Leontin Padurariu, Fereshteh Falah Chamasemani, Roland Brunner and Liliana Mitoseriu is an independent scientific work characterized by novelty and originality.
The authors touched upon an important topic related to the evolution of soft inclusions in ceramic materials.
Today, this topic is extremely relevant due to the active search for new composite materials that are suitable for the energy industry.
However, I would like to ask the following questions:
1. Can the polymer used have such a broad generalization as "The present results are general irrespective of the nature of the ceramic powders and of the soft polymeric inclusions generating porosity."? Proposition 467. Many polymers have non-standard Young's modulus and stretch coefficients.
2. The authors consider polymer inclusions in the green body. During the synthesis of ceramics, it is understood that such inclusions "burn out". Combustion products are often able to form reducing conditions, which ultimately can lead to the appearance of new phase formations at the polymer-ceramic interface.
3. There is a widespread opinion that the occurrence of stresses is a parasitic effect. Stresses are the places with the least resistance to fracture. In this regard, I think that it would be useful for the authors to discuss not only the orientation of pores relative to the directions of applied pressure, but also the total orientation of stresses in volume.
Reviewer 4 Report
Dear authors,
Your paper had an interesting aim, but I do not believe you used an adequate type of numerical modelling. With such 2D simplification, an statistical approach is extremely necessary, since your initial microstructure can bias your result. Another weak point in the model, in my point of view, was the material model selected for the powder. With such material model you do not capture the complex deformation behavior of powders, which can invalidate your results when comparing to reality. You suggested using Drucker-Prager model to model the powder, and I agree with your suggestion.
Unfortunately, I believe the paper needs further improvement in order to be accepted.
Round 2
Reviewer 4 Report
With the additions to the text, the reviewer thinks the paper is suitable for acceptance.